# God's Prime Directive: Non-Interference and Why There Is No (Viable) Free Will Defense

David Kyle Johnson 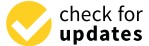

Department of Philosophy, King's College, Wilkes-Barre, PA 18711, USA; davidjohnson@kings.edu

**Abstract:** In a recent book and article, James Sterba has argued that there is no free will defense. It is the purpose of this article to show that, in the most technical sense, he is wrong. There is a version of the free will defense that can solve what Sterba (rightly) takes to be the most interesting and severe version of the logical problem of moral evil. However, I will also argue that, in effect (or, we might say, in practice), Sterba is correct. The only working version of the free will defense requires embracing a view that entails consequences theists traditionally have not and cannot accept. Consequently, the one and only free will solution is not viable. Unless some other solution can be found (Sterba argues there is none), the logical problem of evil, as Sterba understands it, either commits one to atheism, or a version of theism that practically all theists would regard as a heresy.

**Keywords:** logical problem of moral evil; James Sterba; open theism; divine non-interference; divine prime directive; Elif Nur Balci; Janusz Salamon; the free will defense; Alvin Plantinga

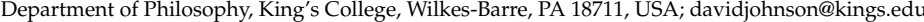

> "I observe all that transpires here, but I do not, cannot, will not interfere".
>
> —The Watcher
>
> *What if . . .* (Episode 1, Marvel/Disney+)

## 1. Introduction

The free will defense only works as a solution to the problem of moral evil if one embraces a libertarian understanding of free will. If compatibilism is true, and a person can act freely even if they are causally determined to act as they will, then free will cannot explain why God allows evil. On compatibilism, "to stop that evil God would have had to violate someone's free will" cannot be used as an excuse for why God allowed an evil because, on compatibilism, any evil could have been stopped by God (without violating free will) simply determining the agent to freely choose differently than they did. For example, on Frankfurtian compatibilism—which suggests an action is free if it done in accordance with one's second-order desires (see Frankfurt 1971)—God could have prevented the holocaust, without interfering in Hitler's free will, by simply giving (determining for) Hitler different second order desires.

To be sure, some have tried to defend a compatibilist version of the free will defense (see Almeida 2017; Gillett 2018). However, I take it to be widely regarded by most philosophers that such efforts are doomed to fail.[1] Indeed, Alvin Plantinga—whose free will defense is most famous—explicitly stated that the free will defense must assume an incompatibilist view of free will (see Plantinga 1985, p. 45). It is for this reason that, in this essay in which I will be exploring whether there is a viable version of the free will solution to the logical problem of moral evil, I shall assume that it requires a libertarian understanding of free will.

In a recent book and article, James Sterba has argued that there is no free will defense (see Sterba 2019, 2020). It is the purpose of this article to show that, in the most technical sense, he is wrong. There is a version of the free will defense that can solve what Sterba (rightly) takes to be the most interesting and severe version of the logical problem of moral

evil. However, I will also argue that, in effect (or, we might say, *in practice*) Sterba is correct. The only working version of the free will defense requires embracing a view that entails consequences theists traditionally have not and cannot accept. Consequently, the one and only free will solution is not viable. Unless some other solution can be found (Sterba argues there is none), the logical problem of evil, as Sterba understands it, either commits one to atheism, or a version of theism that practically all theists would regard as a heresy.

## 2. Sterba's Problem of Evil

The strictest understanding of logical problem of moral evil—as it was articulated, for example, by Mackie (1982)—has always been somewhat uninteresting because it "overplays its hand" (if you will). It suggests that the existence of a tri-omni (all-good/powerful/knowing) deity is logically incompatible with existence of evil such that, if this traditional deity of theism (i.e., God) exists, there should be no evil at all. To defeat his argument, it seems one simply needs to challenge the necessity of the principles on which Mackie bases his argument, or just provide a single logically possible scenario in which God and a single evil event co-exist.

However, imagining such a scenario does not address the concern I have when I think about the logical problem of moral evil. I wonder, not how God could allow any evil, but how God could allow the evil that exists in our world. Especially troubling are particularly horrendous evils, that are either inexcusable (that cannot be justified regardless of their consequences) or from which it seems no moral justifying consequences do or even could come. The holocaust, the Rwandan genocide, the rape and murder of small children—it does not seem that the existence of any of these evils is logically compatible with the existence of a tri-omni God—because if a tri-omni God existed, he would prevent them—and yet they occur. Combine them all together into one possible world, as they are in ours, and it seems obvious that such a world is not a world that a tri-omni being would actualize. While a world with no-evil might not be logically possible, clearly a world with much less evil than our own is (at the least, it would not contain horrendous evil); and the actualization of such a world is what a tri-omni being, by definition, would prefer.

According to James Sterba (2019, 2020), while Alvin Plantinga's famous "free will defense" was initially thought to be the agreed upon solution to the logical problem of moral evil, further debate on the issue revealed that it only functioned as a solution to a less interesting version of the problem. For example, the debate revealed issues with Plantinga's idea that all free creatures suffer from transworld depravity (so that, in every possible world that God could create, all free creatures perform some evil actions). "Faced with such dissension", Sterba points out,

> "Plantinga has entertained . . . that all he really needs to counter the argument from evil posed by John Mackie is simply to espouse the One Wrong Thesis (OW), which just claims that if God tried to actualize a morally perfect world, at least one person he creates would act wrongly. (Sterba 2019, p. 25)

However, since our world is clearly not a world where just one person acts wrongly, Plantinga's solution is uninteresting. It does not address the issue of whether the horrendous evil that actually exists in our world is logically compatible with the traditional god of theism—and that is the harder problem.

However, even Plantinga's original "transworld depravity" solution is inadequate to the relevant task. The fact it's possible that, in every possible world, every free creature does some evil action does not explain why God allows our world to contain horrendous evils. Indeed, Sterba argues that all free will solutions to this problem ultimately fail. To oversimplify a bit, given that God is all-good, he should prevent persons from doing actions that significantly curtail the free will of others—either by God preventing the person from being able to carry out the evil action that they intend to do, or allowing the evil action but keeping any free will restricting consequences from being realized. Since our world clearly includes such evils and their consequences, the existence of our world is not logically compatible with God's existence.

To my eyes, there are a few ways to defend this, all of which I believe Sterba appeals to in one way or another. The most direct way, which Sterba articulates clearly in his 2020 paper on the topic, is by appealing to the Pauline Principle, which suggests that one should never do evil so that good may come of it. Allowing someone to curtail the freedom of another, when one could easily prevent it, is an evil; and even though doing so might accomplish some other good—like the preservation of the offender's free will, or the opportunity to show compassion to the victim—if the Pauline principle is right, such a good cannot be used to excuse the allowance of that evil. To put Sterba's argument more precisely, the Pauline Principle entails three minimal moral requirements—and if they are right, such a good cannot be used to excuse the allowance of that evil. However, since, as Sterba argues, these principles are acceptable to consequentialists, non-consequentialists, atheists, and theists alike, the free-will defense is not an adequate response to the existence of the horrendous free-will restricting evil that exists in our world.

Second, the right of someone (A) to rob someone else (B) of their free will (e.g., by A killing B) is not as morally important as the right to free will that the other person (B) has. Since, if A intends to kill B, an infringement of free will is inevitable, God should act to ensure that the least objectionable infringement of free will occurs. In other words, since restricting the free will of a killer is less morally objectionable than the free will restriction the killer's action will bring about, God should act to ensure that the free will restricting action of the killer does not take place. Or, if God will not stop the decision, God should ensure that the consequences of the killer's intended actions are not felt—especially since he could do so without a major infringement of the killer's free will (by, say, giving him a flat tire that derails his plans).

A third, related, way to defend this idea is to point out that if the existence of free will is the greater good that excuses God's allowance of evil—which is what the free-will defense seems to necessarily imply—then God cannot allow actions that restrict the free will of others. Evils that do not restrict free will would be excusable; but acts that restrict the free will of others—especially the free will of many others –cannot be tolerated. They would reduce the overall amount of free will in the world, and thus the excuse that "God wants free will to exist" could not be invoked to explain why God allows them. If free will is a greater good, then the free will of one person should be restricted if doing so protects the free will of others.

### 3. A Possible Freewill Solution

It is with this last argument in mind that I would like to propose a version of the free will solution (to Sterba's more interesting version of the logical problem of moral evil) that can defend the notion of God's existence. My argument, however, should not be mistaken as an argument for theism. Despite the fact that the solution I will propose is a defense of the traditional tri-omni god of theism, it is not a solution that theists would traditionally (i.e., usually, historically, typically) be willing (or perhaps even could be willing) to accept. Thus the crux of my argument is that the only solution to the relevant logical problem of moral evil is one that is, ultimately, incompatible with (or unpalatable to) theism. It is not viable and thus theism should be rejected.

The central idea of the solution is to insist that, when it comes to the freely-willed actions of free creatures, God must maintain an absolute non-interference policy—a kind of Divine Prime Directive (DPD), if you will. In *Star Trek*, the crew of the Enterprise (or any Federation starship) is bound by "General Order 1", also known as The Prime Directive, which forbids them from interfering in the development of any primitive (i.e., pre-warp) civilization. Even if a planet's inhabitants are enslaving half their population, even if they are about to commit genocide, even if they are dealing out unjust punishments, even if they are about to destroy themselves or be destroyed (say by natural forces)—the Prime Directive demands that the Starship's crew just observe and not interfere. That is not to say that Federation crews do not routinely ignore The Prime Directive. (Additionally, that is also not to say that they should not, although Picard's excuses were usually better than

Kirk's (see Johnson 2015). However, this is what The Prime Directive demands: absolute non-interference.

If God is bound by a similar, divine, unimpeachable, version of The Prime Directive—so that no matter how good or evil a person's action is or will be, and no matter what the consequences of that action are or will be, God must not interfere in any way with their ability to freely perform and bring about the consequences of that action—then the answer to the logical problem of moral evil is obvious. God does not prevent horrendous moral evils because he is bound by a Divine Prime Directive (DPD) which entails that he cannot—and that would include evil actions that restrict or interfere with the free will of others. To work, this solution must insist that (to borrow Sterba's example) even preventing a murder by giving someone a flat tire, or (to borrow my own example) preventing the holocaust by giving young Hitler a heart attack, are off the table (see Johnson 2022). The argument would be that such actions rob the actor of the opportunity to freely choose to do the relevant actions and make their effects felt, and thus constitute violations of free will. However, if this view was embraced, it would explain why God did not and does not take such actions.

In my assessment, I am not alone. In reply to Sterba, Salamon (2021) argues that

"Sterba's recent restatement of the logical problem of evil overlooks a plausible theistic interpretation of the divine–human relation, which allows for a theodicy impervious to his atheological argument, which boils down to God's failure to meet Sterba's "Evil Prevention Requirements". I argue that such requirements need not apply to God in a world under full human sovereignty, *which presupposes that God never intervenes to change the natural course of events to prevent evils . . . "*. (Salamon 2021, p. 1 *emphasis added*)

I will return to Salamon's argument later.

An initial difficulty with this suggestion might seem obvious: how can a tri-omni entity be bound by a directive? The answer is obvious too, but also problematic. What's obvious is that the directive must be self-imposed. God recognizes that he should be bound by it—that non-interference is always best—and thus never violates it. He is thus not controlled or limited by some outside force; God's non-interference is just a consequence of God being the best possible being. What's problematic is, the notion that "non-interference is always best" is difficult to defend.

Indeed, in *Star Trek*, it is very obvious—quite often—that non-interference is not always the morally best policy. Genocide should be stopped, slavery should be abolished, unjust punishments should not be rendered, species-ending natural calamities should be averted. Such things are morally preferable. Likewise, it seems equally obvious that preventing murders and holocausts with flat tires and heart attacks is morally preferable. The same is true for preventing person A from unjustly restricting the free will of person B, even if it requires interfering in the free will of person A. As Sterba points out, this is why societies have laws that prevent such behaviors. We view the right to free will action as more paramount than the right of others to restrict it, and thus preserving the former is morally preferable.

The theist might attempt to defend the DPD in the same way that the federation defends The Prime Directive in *Star Trek*: we cannot see all ends, and thus cannot know what the ultimate consequences of interference will be. They might be worse, and thus we should just play it safe. (For example, perhaps someone on the planet that Picard is about to save will grow up to be the next Hitler, or Khan Noonien Singh). As a defense of *Star Trek*'s Prime Directive, this argument is problematic but perhaps defensible (see Johnson 2018). However, it decidedly cannot be used by traditional theists, who believe that God has perfect knowledge, to defend the DPD. If God has perfect knowledge, he *can* see all ends.[2]

Of course, this raises the specter of suggesting that, because God can see all ends, God knows that, despite how it appears to us, non-interference is always the best policy (i.e., that non-interference will always lead to the best results). This, however, is far too convenient.

Non-interference *always* leads to the better consequences? That is just special pleading. It cannot be that allowing evil will always naturally result in more good. Sometimes it would and sometimes it would not. Of course we do not know exactly how all the "would"s and "would not"s are distributed among the list of all possible evils—which evils would be made better by interference and which would not—but there are a countless number of ways they *could* be distributed. To suppose, without argument, that they happen to have the one and only distribution they must in order for God to exist—where everything is a "wouldn't"—smacks of desperation. "God doesn't stop evil because it just so happens that he never should" is not a real solution.

Along these same lines, however, one might insist that God does interfere when it is best, and does not when it is not—and it only seems worse to us when he does not interfere because God sees ends that we cannot see. However, this suggestion abandons the DPD under consideration in favor of skeptical theism—which is not a free-will solution and also fails as a response to the moral problem of evil because of its own faults (see Sterba 2019, chp. 4; see also, Johnson 2013, 2021). So, we still do not have what we need, which is a reason why God would embrace a non-interference policy.

## 4. Justice Is No Excuse

In "A Modified Free-will Defense", Balci (2022) argues that God may not interfere to prevent evils that hinder the free will of others because to do so would circumvent God's desire for justice. Although Balci does not use the term Prime Directive, or even "non-interference policy", she defends the same idea and incorporates it into what she calls her "structural free-will defense".

> I suggest that this divine permission should be understood as structural permis-
> sion. . . . I argue that, rather than focusing on individual acts and their results, we
> should focus on the structure in which free-will operates. Accordingly, structural
> free-will refers to a standard and general structure in which all human actions
> occur. . . . The idea advocates the free-will as a possibility for all human beings,
> but not the free-will distributed among individuals. Therefore, issues such as
> how this possibility is realized through human actions and what proportion and
> amount of evil these actions cause become irrelevant to the structural free-will
> defense. . . . In the structural free-will defense, there is no distinction between
> the person who uses their free-will and the person harmed by this action. Evil is
> only related to structural free-will as a general possibility . . . . In the structural
> defense of free-will, God is the creator of this structure. Just as structural free-will
> has nothing to do with individual moral free-will, God acts in harmony with this
> structure as the creator of it. In other words, God, the creator of this structure,
> cannot be understood as one who dispenses free will to each individual for use in
> each particular action, thus openly permitting evil acts. God does not prefer one's
> freedom to another's freedom. God does not give somebody more significant
> freedom and deprive others of it. God allows individual human actions, whether
> good or bad, to take place and this permission should be understood as general
> permission. God has revealed this structure in a way that guarantees the free
> action of everyone. . . . Therefore, God cannot be held responsible for the moral
> consequences of any evil act of an individual who uses the freedom provided
> by this structure. How the individuals use this freedom is entirely up to them.
> (pp. 4–5)

She also rightly observes that "the moral justification for why God created such a structure and why God does not structurally interfere with human choices cannot be explained by structural free-will defense theory alone". (p. 5) In other words, suggesting God *does* abide by a non-interference policy does not explain *why* he abides by a non-interference policy—and a successful defense would need to explain why. Balci, however, has an answer. She suggests God does so to promote justice. "[T]he promotion of free will

[is] not [for] the promotion of amount and distribution of free will itself but as the promotion of the principle of justice. . . . [it is] a free-will defense in which justice is promoted". (p. 5)

How does free will promote justice? Balci tells us.

The fact that [hu]man[s] can be justly held responsible by God is only possible if God gives [hu]man[s] free-will. Since humans are held responsible for their free will, they are the creator of their own actions, whether significant or insignificant, good or bad. The human under the divine sovereignty is an agent being morally tested by God. . . . Whomever God is testing, is sure that they will receive righteous rewards and punishments from God, due to their relationship with the just God. Therefore, in this relationship, God is also morally responsible to give just responses to His servants. . . . If free-will is to be promoted, this is only possible [if] the moral relationship between human's accountability and God's just responses is not violated. . . . A God who interferes with human's [sic] free will cannot continue to hold humans accountable in a just way. If God occasionally intervenes in human's [sic] actions, there would be no point in divine condemnation . . . an action that will be the subject of divine moral judgment must be an act of complete free will and uninterrupted. (pp. 6–7)

In other words, God maintains an absolute non-interference policy when it comes to human free action because, otherwise, he will not be able to ensure justice—he will not be able to reward or punish humans for said actions. The problems with Balci's argument are two-fold.

First, it's quite clear that the rewards, and especially the punishments, that Balci has in mind do not accomplish justice. Balci's writing makes it clear that she adopts very traditional religious views, which would include the traditional notion of hell, where those who do evil in this world are punished eternally for it. However, of course, no evil that a person does in this world deserves infinite punishment because the crime and the harm it generates are finite—and it is in proportion to such things that "just punishments" are determined. Infinite punishment for finite crimes is not justice, it's overkill. Johnathan Edwards, of course, argued that since sins are against God, and God is infinite, they deserve infinite punishment—but multiple scholars have shown, in detail, why this is essentially a poor ad hoc "just so" rationalization for an invented and ultimately indefensible theological doctrine.[3] Seymour (1998), for example, explains why the arguments of Augustine, Aquinas, and Edwards all fail as defenses of the traditional notion of Hell as a place of eternal torment, and instead argues that the only way to defend such a notion is by modifying it. Hell, if it is to be just, must be a place where people have the free will to continually sin, over and over, for all eternity—thus continually deserving, forever, more and more punishment for what they have done/keep doing. However, not only is this completely contrary to the view of hell upheld by traditional religious believers such as Balci, and indeed the vast majority of monotheists (Christian, Jewish, and Muslim),[4] this conception of Hell leads us to the second problem of Balci's view.

While this modified version of Hell—filled with sinners who just keep sinning and thus deserve more and more punishment—is, by itself, logically possible, it does not seem to be the kind of place that God would (or, given that he is all-good, logically could) allow to exist. What would be the point? "We need a place filled with people who just keep doing evil so I can continue to give them what they deserve for doing evil". Even if they just keep visiting punishments upon each other, this is not the kind of place that even a minimally decent being would allow to exist (if they could prevent it). Allowing evil simply for the sake of punishing evil-doers is not a noble characteristic. While I grant that persons with unreformable characters are logically possible, and that heaven could not be heaven if such persons were included in it, if God were all-good, he would not create a place where such persons could simply continue to sin and be punished for it. Their annihilation, for example, would obviously be morally preferable.

Something similar is true of allowing evil in this world for the sake of punishing evil-doers. This is not something that an all-good being can do. It not only violates the

Pauline Principle—by allowing an evil to accomplish a good—and all three of the moral principles that it entails, but it seems especially cruel. Even if a world in which a murder is punished is better than a world in which the murderer gets off "scott free", no morally decent person would allow a murder so that the murder could be justly punished. However, suggesting that God maintains a non-interference policy because, otherwise, he could not accomplish justice by holding people accountable for what they do, entails that God does exactly that.

To be fair, Balci insists that this is not what God does on her view.

> I can justify this non-intervention through the moral relationship between God and human[s] which appears only in theism. However, this should not be understood as a condition/cause of God's own justice. Otherwise, we might end up with an interpretation in which we accuse God of allowing evil only to do his justice. Free will (non-intervention) and justice (intervention) are not in causal relations. Rather they are two complementary graces in God's relations with human beings. (p. 8)

However, the problem I have articulated here is not one of cause and effect. My argument is not that God's justice causes evil, or that free will causes justice, or vise-versa, or anything of the sort. So pointing out that free will and justice are not in causal relation is irrelevant. The problem is, regardless of what Balci's own personal view of God is, her argument *entails* that God does not ever interfere in human free actions because otherwise holding people morally responsible for their actions is impossible—and that means that *the reason* God does not step in to prevent evil is so that he can punish evildoers.

We can better understand the problem by articulating and responding to a footnote Balci added on the topic.

> God's justice can be understood in two ways. The first is that God gives a reward to the victims of free will He created. Second, it is human's [sic] responsibility how to use free will, and therefore God rewards those who are aggrieved as a result of human action. Reward and punishment, which are God's justice, are the result of free will, something God created. Let us imagine, human beings might not have used their free will for evil. In this case, God's punishment ceases to be an inevitable result. So, punishment is only an option. As Keith Ward wisely points out . . . we can consistently think that God creates the possibilities of evils without wanting actual evils to happen. (Footnote 9, p. 12)

There are a number of issues here. First, there is no difference between the two understandings of God's justice Balci articulates. "Giving a reward to the victims of free will" and "rewarding those who are aggrieved as a result of human action" are the exact same thing. Second, "rewarding the victims" of free acts is also morally problematic as a reason for allowing an easily preventable evil. No one should allow a rape so that they can reward the victim of that rape. Third, rewarding the victims of evil is not the only way to understand justice; it is not even the primary way justice is understood. Throughout the paper, it's clear that rewarding good action and punishing evil action are the main aspects of justice that Balci has in mind. Fourth, pointing out that human beings might not have used their free will for evil is completely beside the point. Since God knew that they *could* use it for evil, if God knows he will never interfere in free will for the sake of justice, God is knowingly willing to allow evil for the sake of punishment—and that is not something even a morally decent being would be willing to do. Furthermore, on Balci's theistic view, God clearly knew humans would use free will for evil, so the possibility that they might not is moot.

To be charitable, one might understand Balci as suggesting that God does not adopt a non-interference policy *explicitly* for the sake of punishing evil doers, but instead simply for the sake of being able to give people what they deserve—whether it be reward for doing good or punishment for doing evil. After all, that is essentially the definition of justice. The problem is, the adoption of a non-interference policy (or Balci's structural free will

approach) is not necessary for the "reward the good" part of that equation. God could grant all humans free will and decide to not interfere when people will freely choose to do good (thus enabling him to reward them), but then *to* interfere when they freely choose to do evil.[5] The only reason to adopt a *complete* non-interference policy—for God to take Balci's structural approach—is to make possible the punishment of evil; the reward of good is possible without it. So, despite what the "stated" reason is that God adopts a non-interference policy, on Balci's "structural free will/justice" defense, in effect the reason is to punish evil-doers—and, again, that is not something a morally decent being would do.

However, this brings to mind one final possibility: What if interfering with one kind of action is not possible without interfering with the other? Justice would still fail as a reason to adopt a non-interference policy—again, as I am sure Sterba would point out, doing so violates the Pauline Principle and its three implied moral principles. However, it is to the potential impossibility of (what we might call) "compartmentalized interference" that I shall now turn.

## 5. Motivating the Divine Prime Directive

For reasons that Sterba has already pointed out, we cannot defend the DPD by simply saying that "free will is just that important"—so important that no violation can be tolerated. If it's that important, it should be maximized, and maximization will require occasional violations (e.g., to prevent one person using their free will to violate the free will of ten others). This would be like thinking political liberty is of maximal value but then advocating for anarchy. Philosophers from Locke, to Mill, to Rawls all valued liberty, but also recognized that it must be occasionally restricted in order for it to be maximized. Popper ([1945] 2012) recognized something similar about tolerance. Tolerating everything would tolerate intolerance, so those who love tolerance must limit it. So valuing tolerance means you cannot tolerate intolerant views, valuing liberty means you cannot tolerate that which limits it, and the same is true of free will. Valuing free will means you cannot tolerate decisions that restrict or limit it in others.

Where we might start to find a viable defense of the DPD is in the very nature of libertarian free will. Consider Frankfurt-style counterexamples, which are supposed to negate libertarian definitions of free will by showing that free will does not require alternate possibilities (see Frankfurt 1969). In them, someone has a device in their brain that monitors their brain activity such that, if they are about to fail to freely decide to do X, the device kicks in and makes them do X. However, if they are about to freely choose to do X on their own, the device only monitors their brain activity and lets them do it. Frankfurt argues that if the latter occurs, clearly the person acts freely—and yet they cannot do otherwise (because, if they were about to do otherwise, the device would kick in and prevent them from doing so).

A vast literature exists on the topic, which I will not summarize here—except to point out three things: (1) I think such counterexamples fail to falsify the libertarian definition (see Johnson 2016, lecture 18); (2) if they do succeed, there is no free will solution (because, as noted above, it only works on a libertarian understanding of free will); and (3) one common libertarian response is that, if libertarian free will exists, such a device is impossible. For the device to kick-in, at least some "flicker" of a free will decision must have already occurred—enough to ground moral responsibility and say that the decision was made (see Speak 2002). In the same way, one might argue that, given the nature of free-willed decisions, it's not possible for even God to know whether someone is about to freely decide to do good (so he knows not to interfere), or about freely decide to do evil (so he knows he should interfere). For the same reason that a device that only kicks in when you are about to "fail to decide to do X" is impossible, God cannot decide to only interfere when a person is about to freely choose to do evil. In order for free-will decisions to be possible, God simply must not ever interfere, and let us act as we will.

This cannot be the whole story of the solution, however; even if this is true, why does not God step in after evil actions are freely chosen to make sure that their severe (e.g.,

free-will restricting) consequences are not felt? To answer this question, one might appeal to free will's fragility. Perhaps free will is so delicate and unstable that no violations, even of its consequences, can be tolerated. If you violate it, *at all*, in any way, it ceases to exist. Even if you give someone a heart attack a day before they might freely choose to do evil, that is too much. One way to defend this idea would be to suggest that, if God starts interfering in free will, using his divine powers to prevent or curb the consequences of freely chosen evil actions, it will become quickly apparent that such actions are not "really possible" and thus the ability to freely choose to do them will be removed. In short, in a world in which God always acts to prevent or curb the consequences of horrendously evil acts, "the ability to choose to do otherwise" is not robust enough to ground libertarian free will.

Whether this is true is difficult to determine. On the one hand, persons and governments can interfere with free will without eliminating it; on the other hand, they are not using unlimited divine powers. If things always worked out so that, coincidentally or magically, evil actions were either always stopped or limited to non-free will violating consequences, people might stop trying. Regardless, it seems that it is at least a defensible view.

Salamon (2021) defends the idea that "God never intervenes to change the natural course of events to prevent evils" by appealing, not to ensuring justice, but simply to the greater good of "respecting human sovereignty". (p. 1) He invokes Giovanni Pico della Mirandola's doctrine of "collective selfhood", which he also takes to be a part of Dostoyevsky's solution to the problem of evil, which suggests that God relates to humanity as a whole, rather than individually. Consequently, God grants *humanity* (as a whole) free will—not individual persons—and thus necessarily never directly interferes in human actions, good or evil. Although I will return to the overall viability of Salamon's view later, I take this to be another possible defense of the DPD.[6]

Regardless of which option you take for defending the idea that God adopts an absolute policy of non-interference, we can now see how this solution answers Sterba's argument specifically. He admits that there are certain exceptions to the Pauline Principle. For example, if a person is forced to choose between killing one native themselves, or allowing 20 (or 200, or 2000) to be killed, the evil in question is justified. The solution in question suggests that God is forced, despite his omnipotence, to choose between there being no free will (and thus no moral good) at all, and allowing there to be whatever moral good and evil we choose to create. The suggestion here is that God is morally justified in risking the latter in the name of avoiding the former because it is a greater good.

The solution becomes unviable, however, when its consequences are considered.

## 6. Considering the Consequences

First of all, embracing the idea that God binds himself to an absolute non-interference policy is almost tantamount to embracing deism—deism being the doctrine that God created the universe but does not interfere in its operations—and most theists openly reject deism. There are, however, a few subtle differences between deism and the view being considered. Deists are traditionally more concerned with violations of natural law (not free will); deists, such as Thomas Jefferson, denied the existence of miracles. However, embracing the DPD would not mean that miracles could *never* occur—just not ones that interfere with free will (although, it should be noted, most miracles that theists believe in would interfere with free will). Another problem resides in the fact that the god of Deism is often likened to a watchmaker, who designed (or wound up) the universe and then let it go, knowing that everything that happened would be according to his design. Such a universe is a deterministic universe in which libertarian free will cannot exist; thus, on this view, no version of the free-will defense is viable. Another difference is that deists usually maintain that God is unconcerned with humanity and the goings on of the universe—but God embracing a non-interference policy does not entail that he is unconcerned.

The theological view most compatible with God embracing the DPD is Open Theism. On Open Theism, to create our universe, God did not actualize an entire possible world

(complete with our entire history, past, present, and future). He started it in such a way that it would eventually have free creatures, and then followed along with its development, in real time. On Open Theism, God is in time, not timeless, and does not have knowledge of the future. So he does not know what we will choose or what the subsequent consequences of our choices will be. (For some open theists, he *cannot* have knowledge of the future—although, they argue, that does not limit his omniscience (see Rhoda et al. 2006; Tuggy 2007).) However, on Open Theism, he can and does still care about what choices we make, and so is concerned about the fate of humanity.

To be clear, I am not saying that Open Theism is committed to the idea that God binds himself to the DPD. Even without foreknowledge, God could occasionally know that some person is about to freely choose to do something evil and act to prevent it. After all, we can do this. What I am saying is that it's easy to fit the DPD into Open Theism. God starts the universe, gives humans free will knowing that evil will likely result, but cannot foresee the exact consequences of free will decisions. This not only means that he does not know whether or not the amount of good in the world will outweigh the evil, so that the "risk of evil made inevitable by free will" will ultimately be "worth it"; but it also means he would not know the consequences of his own acts, like if he were to choose to limit certain freely willed decisions or their consequences. Interfering with free will, even once, could endanger its existence—and so, either because he still thinks it is worth the risk, or (ala Salamon) he sees humanity's sovereignty as an overall justifying good, God embraces a non-interference policy and allows human history to be solely a consequence of what we freely choose.[7]

However, although the DPD/Open Theism view seems to reconcile God's existence with the evil that actually exists in the world, it does come with consequences—and those consequences are consequences that most theists would not be willing to accept. For example, it means that most of the Bible must be rejected. It is full of stories and doctrines which entail that God has and is willing to interfere in the free will decisions of humans. God commands Abraham to sacrifice his son (and then prevents him from doing so) (Gen. 22), "hardened the heart" of Pharaoh (Exodus 9), and then led Israel out of Egypt, intervened in many battles (e.g., Joshua 10), and controls the decisions of both kings (Proverbs 21:1) and ordinary men (Proverbs 16:9). Additionally, if free will is so fragile that God cannot give someone a flat tire to prevent a murder, he certainly cannot reveal himself to prophets, or incarnate himself, go around performing miracles, and start reforming religion.

Now, to be fair, I could be sympathetic to the argument that incarnation is the only way that God could try to influence human history, and indeed interfere with free will, without endangering it. As a human, he could interfere in free will only as much as any other human could—and since human to human inference in free will does not eliminate it, his actions as a human would not be risky. This does mean, however, that as an incarnate human God would not be able to claim he is divine, or prove to be with miracles; this would put a force behind his interference that, according to this view, would endanger free will. As C.S. Lewis put it, "Merely to override a human will (as His felt presence in any but the faintest and most mitigated degree would certainly do) would be for Him useless. He cannot ravish. He can only woo" (Lewis 2021, Letter #8). So, at best, the DPD/Open Theism view must consider the gospels to be gross exaggerations.

This also means that, on the DPD/Open Theism view, most petitionary prayer is completely useless. Whether it be for someone to "accept Christ", for God to guide the actions of a surgeon, for no one to be hurt at a football game, or for a good parking spot at the mall—fulfilling a petitionary prayer almost always requires God to interfere in the freely willed actions of humans. Indeed, since on this view, flat tires and heart attacks to prevent horrendous evils are off the table, even prayers for God to alter the weather would be off the table. Most miracles certainly would not be permitted—especially those that would make God's presence obvious. At best, one could pray for God to whisper a non-interfering unclear "woo" into someone's ear.

This is all, of course, contrary to what almost all theists traditionally believe. They usually embrace the Bible's teachings (even if they do not consider it inherent), consider the gospels to be at least somewhat historical, believe that miracles have and still do occur, and believe that petitionary prayers are not pointless. However, perhaps the most unpalatable consequence of this view is that human history is completely outside of God's control. If how human history turns out is completely up to us, then there is no guarantee at all that what God wants to happen in history will happen. Not only is this unbiblical (Isaiah 46, Job 42), but it is contrary to the very idea of God's sovereignty which is arguably as much a part of the traditional conception of God as is omniscience, omnipotence, and omni-benevolence.

Now, it might be possible for a temporal view of God that embraces the DPD but that also includes divine foreknowledge to rescue divine sovereignty. On this view, God would consider multiple ways of starting the universe, look into the future of each to see how granting its creatures free will will turn out (what future will they determine for themselves), and then start the universe that has the outcome he wants. However, there are two major problems with this view. One is explaining how divine foreknowledge is compatible with libertarian free will, which I have argued elsewhere is impossible to do (see Johnson 2009). If knowing infallibility beforehand how someone will freely choose is impossible—because that entails that they could not have chosen otherwise—then it is impossible for God to predict what future free will creatures will determine for themselves. The second problem is that this view seems to land us right back into the problem it is trying to solve: reconciling God's existence with the existence of horrendous evil in the world. If God is all good, but knew the universe would turn out like it has, he should have set it up differently. For this view to work, this (i.e., our universe's history) has to be the best possible way human history could have turned out—and that is hard to believe.

## 7. Why the DPD Solution Is Not Viable

It's hard to believe, but of course it's not logically impossible—which is why the DPD solution I have articulated is at least *a* free will solution to the interesting version of the logical problem of moral evil Sterba articulates. In fact, it is the only one. Any free will solution that allows God to occasionally interfere with free will will raise unanswerable moral questions about why he did not in this or that situation—such as to prevent the holocaust or some other such horrendous evil. Now, some might think that one solution is enough to "solve" Sterba's problem, but in all fairness it is not.

Alvin Plantinga has argued that, to answer arguments which suggest that A and B are logically incompatible, one must only tell a single story in which A and B are true together—and the story need not be true or even believable (see Plantinga 1974, p. 58). While it is technically true that such a story demonstrates that A and B are logically compatible, it's not clear this tactic always solves the problem. If it is the *only* story a person can think of—the only way they can imagine that A and B are true together—then, if that person thinks that A and B are true together, it's a story that person must embrace. If the story is absurd, that is a problem. What's more, if it is established that the story in question is the *only way* that A and B could be true together, then everyone who believes A and B must accept the absurd story as true; and, again, that's a problem. Even though it is a story in which both A and B are true, if the story entails things that such persons are not willing to embrace, especially if it conflicts with things that such persons believe because they also believe A or B, then it does not function as a satisfactory answer to how, logically, such persons can believe both A and B—and that, rather than the mere logical compatibility of A and B, is the real issue.

To put it more formally, suppose you believe both A and B, but I have argued that A and B cannot be true together. You reply with a story, S, in which A and B are true together. However, I then observe that, because you believe A, you also believe C; and because you believe B, you also believe D—and I observe that story S is one that entails that both C and D are false. Since story S is not something you can believe, but it is the only explanation

you have offered as to how A and B could be true together, you have not successfully defended your ability to believe both A and B. You could logically believe both A and B if you also believed S—but you do not, so you cannot. For example, in 2011, I argued that the only way to solve the logical problem of natural evil—to logically reconcile God's existence with the fact that the inevitability of natural disasters is woven into the very laws of our universe—is to embrace the idea that someone or something else created our universe (e.g., that we live in a computer simulation; see Johnson 2011). Since theists actively reject the notion that someone or something else besides God created our universe, this solution cannot be used by theists to logically reconcile their belief in God with natural evil.

Likewise, the solution I have proposed here, which tries to answer how God's existence can be reconciled with the moral evil that actually exists in our world, suggests that God, without knowledge of the future, granted his creatures free will, knowing that evil would likely result, and then vowed to adhere to an absolute non-interference policy no matter how bad things got. This, however, is not a solution that theists would traditionally be willing to accept as it is difficult to defend and too contrary to the typical theist's theology and worldview. Not only is such a policy morally problematic, but embracing this view defies scripture, renders petitionary prayers unanswerable, and denies God's sovereignty (doctrines that theists believe primarily because they believe in God). The view is heresy.[8] Consequently, the problem Sterba presents has no *viable* free will solution; the solution I have articulated does not explain how theists can both believe in an all-good god and acknowledge the evil that exists in the world. Additionally, unless some other kind of solution presents itself—which Sterba convincingly argues in his book, it does not—the logical problem of moral evil, as he presents it, provides adequate reason to embrace atheism.

## 8. A Final Remark

In conclusion, let me make clear that, while Salamon (2021) would undoubtedly agree with me that the DPD serves as a solution to Sterba's argument, he would deny my suggestion that it is unviable. While he admits that there is a "need to show that [his] views are broadly compatible with at least some 'traditional' interpretations of theism" (p. 4), he believes he accomplishes this task. To do so, he points out how the views he defends "emerge uncontroversially from Eastern Orthodox Christianity" and lists a number of Eastern Orthodox scholars who embrace these kinds of views and ground them in "Byzantine patristic sources". (p. 4)

To this, my reply is as follows: Like Salamon, I too know scholars—in my case they are catholic and protestant—who embrace the kind of non-interventionist Open Theism, and all that goes with it, necessary to make this solution work. Additionally, to do so, they quote scripture and ancient church founders. However, that does not mean they define what the view of the church, the vast majority of believers, or what theists traditionally believe. After all, even the most famous open theists admitted in the very title of their book—*The Openness of God: A Biblical Challenge to the Traditional Understanding of God*– that, although their defense of Open Theism was biblical, it was a challenge to the *traditional* view (see Pinnock et al. 1994). Indeed, when most believers find out what open theism is—including pastors, church leaders, and those most interested in doing apologetics—they object vociferously and label open theists unorthodox heretics. For example, the vote to remove open theists Pinnock and Sanders as members of the Evangelical Theological Society only failed procedurally because it required more than a majority vote. (It was just short of the 2/3 necessary to pass; see Robinson 2014).

I contend that the same would happen to Eastern orthodox scholars that openly embrace Salamon's view and admit what it entails. Indeed, in his defense against the charge of deism, not only does Salamon admit that non-interference in not "traditional", (p. 12), but he admits that it entails that the only way that God can "engage with and inspire human beings in the course of history" is by being "present to human consciousness". (p. 12) This means no direct revelation, no incarnation, and (virtually) no miracles. (He

might deny that what he says here implies it is the *only* way, but like I pointed out above, non-interference entails that such things cannot happen.) Obviously, this is not acceptable to most theists.[9]

Likewise, he also admits that the sense of communal responsibility to which he appeals to defend his argument entails that "precise attribution of responsibility for particular evils [is] impossible (making all evil essentially social)". (p. 13) That would mean that no individual can be held responsible for their sins—and that completely invalidates both the doctrine of penal substitution and of hell (and heaven for that matter), and the very idea that *individuals* need to be forgiven of *their* sins (which I take to be the defining characteristic of all versions of Christianity, and to be essential to monotheism in general). (Notice that, if God grants free will to humanity as a whole, and not individual humans, it also makes no sense to punish individuals for individual sins, and Balci's whole notion of justice is out the window.)

I have no doubt that Salamon knows of scholars who argue for the acceptability of embracing such views, and do so by citing orthodox scholars. Undoubtably, those scholars would argue that theists *should* abandon the views that they have traditionally held and, instead, embrace their view of god; and, undoubtably, that is what theists should do if they want a version of theism that is defensible. However, this in no way makes such views traditional, and thus in no way makes his solution to Sterber's problem viable. To put it simply, these are views that neither can nor would be embraced by the vast majority of people who do, or ever have, called themselves theists.[10] Indeed, the picture of the universe the solution paints is almost indistinguishable from one in which God does not exist at all.

At best, it paints a universe not unlike the fictional Marvel universe, in which "The Watcher" watches the events of the multiverse—and even cares about what happens—but does not, cannot, and will not interfere. However, of course, The Watcher is not the god of traditional theism.[11]

**Funding:** This research received no external funding.

**Institutional Review Board Statement:** Not applicable.

**Informed Consent Statement:** Not applicable.

**Data Availability Statement:** Not applicable.

**Conflicts of Interest:** The author declares no conflict of interest.

## Notes

[1]  Almeida (2017) argues that, in order to solve (what I shall call the less interesting version of) the logical problem of moral evil, the compatibilist just needs to show that "there is at least one metaphysically possible world in which God coexists with evil". (p. 57) While I am not convinced that he is right, or that such a world is possible on compatibilism, his argument decidedly does not address the more interesting problem that Sterba (2019, 2020) points out. If there is a possible world in which everyone always freely chooses to do the good, which if compatibilism is true there necessarily must be, then such a world is the kind of world God would necessarily create. Even if all possible worlds have some evil, God would still create the world with the least evil—and that decidedly is not our world, since it obviously could have less evil. Thus the existence of our world is logically incompatible with God's existence.

[2]  Before this essay is over, we will consider the possibility that God cannot see all ends.

[3]  This argument never appeared until after the doctrine of hell was adopted, thus it is a "just so" justification for an invented doctrine (not the reason that it arose). For a very thoughtful and through refutation of this argument, see Patheos' "Infinite Punishment for Finite Sins" at https://www.patheos.com/blogs/daylightatheism/essays/infinite-punishment-for-finite-sins (accessed on 14 September 2022).

[4]  The view has a number of problems that make it unpalatable to the traditional theist. A lifelong sinner could stop sinning in the afterlife, and thus no longer deserve punishment. In fact, character reformation would be possible such that the sinner would no longer deserve to be in Hell. What happens then? Do they enter Heaven? Conversely, Heaven would have to have free will as well. What happens to those who sin in Heaven? Not even adding purgatory can deal with these kinds of problems. An afterlife that has these properties is nothing like theists have traditionally described.

[5]    Even if knowing beforehand how someone will freely choose is impossible, God could at least step in after the evil act is done to prevent the consequences of the evil act from being felt. Although such knowledge is not always impossible—sometimes I can know when someone is about to freely choose evil—later, I will articulate how open-theism, which suggests that infallible foreknowledge of the future is impossible for God, plays a role in the only solution that works.

[6]    It's worth noting that Ward (2007) has a similar solution, which suggests that God wants an independent universe with independent beings, and so *almost never* interferes. Engaging with Wards work here would take me too far off topic, but I will mention below (in an endnote) why I think it has the same shortcomings as Salamon's.

[7]    Notice that this makes my proposed solution somewhat like the defense of The Prime Directive that we see in the *Star Trek* Universe—at least in that, like Kirk and Picard, God cannot see all ends.

[8]    The proposed solution most certainly is not compatible with so called "classical theism", which has been considered the orthodox view for hundreds of years; it holds not only that god is tri-omni, but insists that God is outside of time, has complete foreknowledge, and acts in the world in every moment to preserve its existence.

[9]    I believe that something similar is true of Adam Ward's view (which, recall, says that God *almost never* interferes). It gives up divine sovereignty, would have to admit that most miracle stories are fictions, embrace a very ineffectual view of petitionary prayer, could not tolerate incarnation, etc. However, a full exploration of Ward's view would require another paper.

[10]    It certainly is not compatible with "classical theism". See note 8.

[11]    Two things are notable here. (1) Even The Watcher, by the end of the *What if . . .* series, realized that interference is sometimes morally necessary. This makes the view that God never interferes even more difficult to defend. (2) Invoking The Watcher raises the issue of whether invoking a multiverse could solve the problem of evil (see Megill 2011). However, it is not appropriate to discuss this question in depth here because it is not a free-will response and my concern here is only whether there is a viable free-will response (although it is worth noting that Ward (2007) tries to combine the free will and multiverse defense; I will save my comments on that for the hypothetical future paper I might write about his argument). Regardless, the multiverse solution is not a viable solution either because (a) it is a heretical view, (b) Monton (2010) has already explained why such solutions fail, and (c) they raise the problem of no best world, which itself entails that God cannot exist (see Johnson 2014).

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
