# Peer review of "God’s Prime Directive: Non-Interference and Why There Is No (Viable) Free Will Defense"

_religions, doi:10.3390/rel13090871_

Round 1

Reviewer 1 Report

This is a very well researched article/review. There is coherency and critical arguments are substantially developed. The author brings to the table an amazingly broad yet firm foundation of reading. He/she has accurately laid out the ways that this book would make a significant contribution to the scholarship. It promises intelligent and fruitful ways we might come at these crucial questions. Yes, I myself would certainly recommend it to undergraduate and graduate students  and fellow scholars, and so would many others. YES, I strongly recommend publication.

Author Response

Thank you for the kind review and recommendation. 

Reviewer 2 Report

The author presents arguments in favor of the thesis that there is a valid version of the logical argument from evil and that the defense from free will is ultimately unsuccessful. If we accept the author's starting points, his conclusions are valid. The author undoubtedly knows the subject matter and has command of fundamental philosophical tools.

However, one must ask whether the defense of free will is necessary for the validity of what is called "classical theism"? What exactly is meant here by the term "classical theism"? It would be good to further clarify this term. Namely, "classical theism" is a technical term for those contemporary philosophers of religion who continue the classical theistic tradition, especially Thomas Aquinas, and put it in dialogue with contemporary philosophy. Perhaps it would not be bad if the author takes into account those authors (like Brian Davies and Edward Feser) who think in a different way about the relationship between God and evil, moreover, they operate with a less anthropomorphic concept of God that is quite different from the one presented in this article and the existence of whom, we hold, is not necessarily called into question by a logical argument from evil. If he did so, we believe, his argument (and thus the contribution itself) would greatly increase its value.

However, we ultimately leave the decision to the author and the editors.

Author Response

Thanks for the review.

I’m a bit confused because I never use the term “classical theism.” I use the term “traditional theism,” but I very clearly define that as the view that God is tri-omni (omniscient, omnipotent, omni-benevolent), which are the three properties traditionally referred to by the problem of evil.

What this note made me realize, however, is that I didn’t always draw a distinct line between “traditional theism” (as I just defined it above), and what theists traditionally (i.e., typically, usually, historically) believe (e.g., that God answers prayer, performs miracles, is sovereign, etc.). Sometimes I am talking about the former, other times the latter. So I have now gone through the paper and made that clear, mainly by making sure that when I am talking about the former, the word “traditional” comes before “theist” (or “theism”), and when I am talking about the latter, it comes after (e.g., “what theists have traditionally believed,” or something along that line). (I thought about switching to the word “typically,” but when I quote other authors (open theists, Salamon), they use the word “traditional” to refer to the latter concept, so I kept it.)

What’s more, I don’t think that classical theism—as I understand it—could embrace the solution I propose; not only because people who have embraced classical theism have (traditionally!) also embraced the idea that God interferes in the world—but also because it holds that God is outside of time, has foreknowledge, and conserves the existence of the world (which are all denied by my solution … although I could see an argument where, maybe, “conservation” would not entail interference). Maybe the reviewer disagrees, but I feel like that is a whole other paper, and engaging in that issue would take the paper too far of course.

That said, perhaps a case could be made that Davies and Feser’s particular versions of (what they call) classical theism, which emphasize a “less anthropomorphic concept of God”, might be able to be brought in line with the solution I have proposed. But just like with the scholars that Salamon mentions, that does not make their view “traditional” (it does not line up with what theists historically/typically/usually believe). So, while exploring whether Davies and Feser could embrace the DPD solution would also make for an interesting second paper, I don’t think engaging with that question here in this paper would be useful.

All that said, in addition to the revisions mentioned above, I have added two footnotes where I try to briefly make clear why “classical theism,” as I understand it, would not allow one to embrace the solution I have proposed.

Thanks to the reviewer for their time.

Reviewer 3 Report

The article is well structured, discusses recent literature, and answers relevant objections. Given the intended scope of the article, it is acceptable for publication. I do have some objections I invite the author to consider responding to, but I do not require it in order to recommend publication.

The author argues that the kind of free will defence discussed will probably not be accepted by most theists – although acknowledging (correctly, I think) that many scholars might accept it. Nevertheless, there is a difference between discussing what most theists WILL accept and what they SHOULD accept. Put differently; it is a difference between what will be accepted by the majority of theists and what is well argued (for example, in my view, theists should accept that there are many exaggerations in the Bible). How would the author respond to a free will theodicist saying that theists should accept the free will defence being discussed?

The author has a good critique of Elif Nur Balci. Since that is a recent contribution, it is worth criticizing, but otherwise, it would have been better to discuss a better alternative. For example, Keith Ward (in the book Divine Action, and other writings) has a similar free will defence avoiding some of the objections presented. He would say that God wanted to create an independent universe with independent beings (emphasizing thus not merely free will, but a strong form of free will). God might intervene a few times, but not often, and since the world is strongly interconnected, we cannot know when it is possible for God to intervene while maintaining a sufficient level of independence. While God rarely intervenes, petitionary prayer changes the world in which God acts, which makes it not useless (even if one should usually not expect that prayers are answers – but most theists probably do not expect that in any case). The idea is not to think that God has given himself an unbreakable directive, but that God has created a universe where he does not interfere as long as it is on balance good in order to actualize an independent universe, while he would stop the whole project if it turned into a living hell.

Ward even considers combining the free will defence with a multiverse. The author rejects that a multiverse theodicy is a free will theodicy. I would say that a free will theodicy combined with a multiverse theodicy is a free will theodicy explaining why we have this kind and degree of free will on these conditions in this universe, while there could be another universe with another kind or degree of free will on other conditions in another universe. The free will partly explains the goodness of our universe, which explains why it has been created, and the different universes being on balance good, are created because it is good with a manifold of different goods.

These were some possible objections to consider responding to. A few minor issues in the end: Some quotations are not clearly marked as quotations with indentions. This might just be a question of formatting in the version I received. Also: I am used to God being called “temporal” or “in time” when it is presented as an alternative to timelessness. I am not used to the description of God as “timed”. If this is common in parts of the literature that I am less familiar with, “timed” can be kept. Otherwise, the author might consider changing it to “temporal” or “in time”.

Author Response

Thanks to the reviewer for their time.

I’ll respond in four points:  

(1) I made the timed/in time corrections. Thanks for that! I feel like I’ve seen it elsewhere, but I’m not sure so changing it is the easy fix.

(2) Good suggestion about “should.” I added, to the next to last paragraph, a sentence about scholars who would accept the consequences of my solution, and how they undoubtedly would argue that theists SHOULD abandon their traditional views. The word “should” is ambiguous of course, but when simply understood as “That’s what you should do if you what your view of God to be defensible,” they most certainly would argue that—and they would be right. That doesn’t make their view traditional, so my point still stands—and I think discussing it further would take me too far off track—but it is certainly a point worth including in the main text.

(3) I responded to Balci because her paper appears in the same volume; and I don’t feel inclined to cut that part in favor of responding to Ward. Speaking of which…

(4) Ward sounds interesting, but my paper is already over 10,000 words, so I think I should reserve engaging with this work for a potential future paper. That said, I have added a note for how his view is somewhat similar to Salamon’s, another for why I think it faces similar problems as Salamon, and then I also added a bit about how he tries to combine the freewill and multiverse theodicies in his work (to the last footnote). Again, I don’t think I have the room to engage with his argument here—but I will definitely include it in that potential future paper, so I thank you for bringing it to my attention.